# Nutrition and Health in Arab Adolescents (NaHAR): Study protocol for the determination of ethnic-specific body fat and anthropometric cut-offs to identify metabolic syndrome

Lara Nasreddine[1]*, Mohamad Adel Bakir[2], Tareq Al-Ati[3], Abeer Salman Alzaben[4], Rawhieh Barham[5], Nahla Bawazeer[4], Elie-Jacques Fares[1], Kholoud Hammad[2], Pernille Kaestel[6], John J. Reilly[7], Mandy Taktouk[1]

1 Department of Nutrition and Food Sciences, American University of Beirut, Beirut, Lebanon, 2 Department of Radiation Medicine, Atomic Energy Commission, Damascus, Syrian Arab Republic, 3 Food and Nutrition Program, Environment and Life Sciences Research Institute Center, Kuwait Institute for Scientific Research, Safat, Kuwait, 4 Department of Health Sciences, College of Health and Rehabilitation Sciences, Princess Nourah Bint Abdulrahman University, Riyadh, Saudi Arabia, 5 Department of Nutrition, Ministry of Health, Amman, Jordan, 6 Department of Nuclear Sciences and Applications, Division of Human Health, International Atomic Energy Agency, Vienna, Austria, 7 Physical Activity for Health Group, School of Psychological Sciences and Health, University of Strathclyde, Glasgow, Scotland

☯ These authors contributed equally to this work.

* ln10@aub.edu.lb

**Data Availability Statement:** No datasets were generated or analyzed during the current study. All

## Abstract

The prevalence of adolescent obesity in the Middle-East is considered among the highest in the world. Obesity in adolescents is associated with several cardiometabolic abnormalities, the constellation of which is referred to as the metabolic syndrome (MetS). This multi-country cross-sectional study aims to determine the optimal cut-off values for body fat (BF); body mass index (BMI) z-score; waist circumference (WC) percentile, and mid-upper arm circumference (MUAC) for the prediction of MetS among adolescents from Kingdom of Saudi-Arabia (KSA), Kuwait, Jordan, Lebanon and Syria. A secondary objective is to examine the validity of Bioelectrical Impendence Vector Analysis (BIVA) in estimating BF against the deuterium dilution technique (DDL). In each country, a sample of 210 adolescents will be recruited. Data collection will include demographics, socioeconomic, lifestyle and dietary data using a multi-component questionnaire; anthropometric measurements will be obtained and body composition will be assessed using the DDL and BIVA; blood pressure and biochemical assessment will be performed for the identification of the MetS. Receiver operating characteristic analyses will be undertaken to determine optimal cut-off values of BMI, WC, MUAC and BF in identifying those with MetS. Odds ratios (OR) and their respective 95% confidence interval (CI) for the association of the anthropometric measurements with MetS will be computed based on multiple logistic regression analysis models. The Bland and Altman approach will be adopted to compare BIVA against the reference DDL method for the determination of body composition parameters. This study responds to the need for ethnic-specific anthropometric cut-offs for the identification of excess adiposity and

relevant data from this study will be made available upon study completion.

**Funding:** The study is funded by International Atomic Energy Agency (IAEA), and supported through collaboration among ARASIA State Parties under the framework of the IAEA technical cooperation project RAS6094, 'Applying Nuclear Techniques for the Determination of Body Fat and Anthropometric Cutoffs (ARASIA)'. (ARASIA is a Cooperative Agreement for Arab States in Asia for Research, Development and Training related to Nuclear Science and Technology). There are no grants awarded to individual researchers in the respective countries. Instead, each of the countries receive the supplies and deuterium from the IAEA but there are no grants per se.

**Competing interests:** The authors have declared that no competing interests exist.

**Abbreviations:** BF, body fat; BIA, Bioelectrical Impedance Analysis; BIVA, Bioelectrical Impendence Vector Analysis; BMI, body mass index; BP, blood pressure; CI, confidence interval; DDL, deuterium dilution technique; FFM, fat-free mass; FFMI, fat-free mass index; FM, fat mass; FMI, fat mass index; FTIR, Fourier-transform infrared spectroscopy; GSHS, Global School-based Student Health Survey; HDL-C, high-density lipoprotein cholesterol; IAEA, International Atomic Energy Agency; IDF, International Diabetes Federation; IRIS, International Research Integration System; KSA, Kingdom of Saudi Arabia; LDL-C, low-density lipoprotein cholesterol; MetS, metabolic syndrome; MUAC, mid-upper arm circumference; NaHAR, Nutrition and Health in Arab Adolescents; NCDs, non-communicable diseases; OR, odd ratio; ROC, receiver operating characteristic; SD, standard deviation; SPSS, Statistical Analysis Package for Social Sciences; TBW, total body water; TC, total cholesterol; TG, triglycerides; UAE, United Arab Emirates; UK, United Kingdom; WC, waist circumference; WHO, World Health Organization; WHtR, waist-to-height ratio.

associated cardiometabolic risks in the adolescent population. The adoption of the generated cut-offs may assist policy makers, public health professionals and clinical practitioners in providing ethnic-specific preventive and curative strategies tailored to adolescents in the region.

## Introduction

Non-communicable diseases (NCDs), including diabetes, cardiovascular diseases and cancer represent the main cause of death in the Middle-East, accounting for more than 70% of mortality [1,2]. The escalating prevalence of overweight and obesity is recognized as a major risk factor for several NCDs in countries of the region, which are currently undergoing the nutrition transition with its characteristic shifts in diet, lifestyle and body composition [3]. More specifically, the region currently harbors one of the highest burdens of adolescent obesity worldwide [4], with estimates ranging between 14.4% and 29.6% in the United Arab Emirates (UAE), Qatar and Kuwait [5–7]. In fact, adolescents represent the age group that suffers the most from the adoption of a western lifestyle characterized by long hours of television viewing, computer games and heavy reliance on fast food, all of which are key factors affecting dietary habits and obesity levels [8]. Adolescent obesity is a public health concern as it contributes to multiple metabolic risk factors, including insulin resistance, dyslipidemia, and elevated blood pressure (BP), the combination of which is commonly known as the Metabolic Syndrome (MetS) [9–12]. Proper identification and diagnosis of adolescent obesity is therefore crucial for early management and prevention.

The assessment of adiposity in this age group is usually based on anthropometric indices, the most common being the body mass index (BMI) z-score, waist circumference (WC) percentile and waist-to-height ratio (WHtR) [13–15]. However, these cut-offs do not distinguish between increased mass in the form of fat, lean tissue or bone, and hence may result in significant misclassification [16,17]. Since the pathology and morbidity associated with overweight and obesity is driven by excess fat mass, the ideal assessment tool should directly assess adiposity [18,19]. In addition, the interpretation of BMI z-score, WC and WHtR is based on international cut-offs, such as the ones developed by the World Health Organization (WHO). These cut-offs were primarily established using data from western populations. However, there is growing evidence that ethnic variations in body composition and distribution of body fat (BF) may restrict the generalizability of these cut-offs to other ethnic groups [12,20]. For instance, studies conducted among South Asian adolescents showed that the adiposity level that was associated with increased cardiometabolic risk was observed at a lower BMI cut-off value than that proposed for Caucasian youth [21]. Moreover, although a single age-, sex- and ethnic-independent WHtR cut-off value of 0.5 has been proposed [22], a study conducted among Japanese children and adolescents aged 6–14 years old suggested a range varying between 0.41 and 0.44 [13]. While some studies associated %BF values above 25% for males and 30% for females to be associated with increased risk of high BP and lipid abnormalities in adolescents of different ethnicity [23], Taylor et al. suggested a range of 29–35% to be associated with elevated risk of systolic BP and lipid abnormalities among 9–15 year olds in New Zealand [24]. The use of non-population-specific anthropometric cut-offs could result in the misclassification of adiposity, thus compromising the prevention, early identification and management of metabolic abnormalities. The WHO has therefore recommended the development and adoption of anthropometric cut-offs that are population specific [12,25]. In addition, there is emerging evidence that mid-upper arm circumference (MUAC), used historically as an

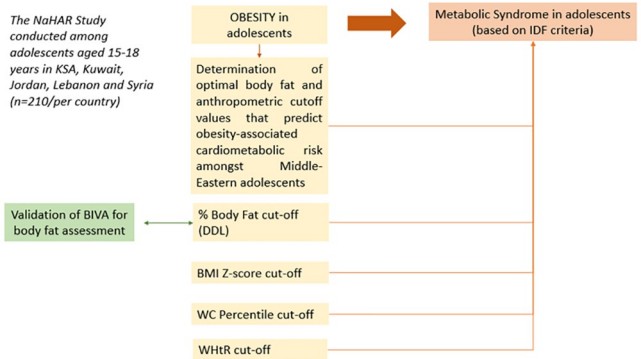

**Fig 1. Conceptualization of the NaHAR study, a multi-country study conducted in the Middle-East for determination of ethnic-specific cut-off values for body fat and anthropometric indicators in adolescents.**
Abbreviations: BIVA: Bioelectrical Impendence Vector Analysis; BMI: Body mass index; DDL: Deuterium dilution technique; IDF: International Diabetes Federation; NaHAR: Nutrition and Health in Arab Adolescents; WC: Waist circumference; WHtR: Waist-to-height ratio.

indicator of underweight, may perform well as an indicator of obesity (high body fatness) in children and adolescents [26].

Current public health surveillance and clinical management of adolescents in the Middle-East depend on the standard methods based on definitions and cut-offs derived largely from populations of European descent. Whether these definitions and cut-offs are optimal for adolescents in the Middle-East is unclear. It is in this context that the multi-country "Nutrition and Health in Arab Adolescents" (NaHAR) study is being launched in several countries of the Middle-East, including Kingdom of Saudi-Arabia (KSA), Kuwait, Jordan, Lebanon and Syria. The objective of this study is to determine the optimal gender-specific cut-off values for BF, BMI z-score, WC percentile, WHtR and MUAC for the prediction of MetS among 210 national adolescents in each of the participating countries (Fig 1). The deuterium dilution technique (DDL), a highly accurate method for body composition assessment, will be used for assessing total body water (TBW), the derivation of fat-free mass (FFM) and fat mass (FM), and hence the determination of %BF [27]. A secondary objective of this study is to examine the applicability and validity of the Bioelectrical Impendence Vector Analysis (BIVA) in estimating BF among adolescents in the region. These assessment methods are practical, inexpensive and relatively easy to use in large field surveys, and have been suggested as valid for the evaluation of body fatness in adolescents of different ethnicities [28,29]. This study will therefore contribute to the generation of validated anthropometric and body fat cut-off points that offer a more accurate alternative to those currently used in the Middle-East for the identification of excess adiposity and associated cardiometabolic risks in adolescents.

## Materials and methods

### Target population

This is a multi-country cross-sectional study that will be conducted in several Middle-Eastern countries, including KSA, Kuwait, Jordan, Lebanon and Syria. In each country, a sample of 210 adolescents aged 15–18 years will be recruited from 4–5 local schools. As this is a methodological study, a non-random purposive sampling approach will be used to recruit adolescent boys and girls across a wide BMI range, to allow for a broad range of body fat in the study sample. Accordingly, obesity will be defined using the widely accepted simple proxies for

**Table 1. Inclusion and exclusion criteria.**

| |
|---|
| *Inclusion criteria; participants should meet all of the following items*: |
| 1. Age 15–18 years |
| 2. Born in the country or having one parent at least born in the country |
| 3. Apparently healthy |
| 4. Having reached the 4th or 5th Tanner stage of puberty (since hormonal changes during puberty may play a role in influencing insulin sensitivity and lipoprotein profile) [34] |
| *Exclusion criteria; participants with any of the following items will be excluded*: |
| 1. Previous diagnosis of inborn errors of intermediary metabolism, or using medications that may alter body composition, blood pressure, glucose or lipid metabolism |
| 2. Having oral illnesses like xerostomia and others that may affect saliva flow |
| 3. Being underweight (defined as BMI < -2 z-score) |

Abbreviations: BMI: Body mass index.

body fatness as BMI > +2 z-scores for sex and age, overweight as +1 < BMI z-scores ≤ +2 and normal weight as BMI ≤ +1 BMI z-scores [30]. The sample size of 210 subjects was calculated using the STEPS Sample Size Calculator and Sampling Spreadsheet [31], using a margin of error of 5%, a confidence interval (CI) of 95% and a prevalence of MetS among adolescents of 10% [32,33]. The design effect was considered to account for the stratification based on BMI (normal weight, overweight, obesity). The inclusion and exclusion criteria are summarized in Table 1.

## Pubertal stage assessment

Acknowledging that pubertal stage may have an impact on body fat levels, FFM accretion, and insulin resistance [35], participants at the 4th or 5th Tanner stage of puberty will be recruited. Pubertal Tanner stages 4 and 5 will be ascertained based on questions as per the method by Carskadon and Acebo 1993 [36]. Questions common for boys and girls inquire about: 1) growth in height, 2) growth of body hair, 3) skin changes, especially pimples. Additional questions for boys ask about the deepening of voice and the growth of facial hair, while those for girls ask about growth of breasts and menstruation (Table 2).

## Data collection

Research teams from each country will participate in standardized training sessions for the harmonization of data collection procedures and protocols. In each country, upon ethical

**Table 2. Scoring* and classification of the Tanner stage.**

| Correspondence to Tanner staging | Boys (add points relative to voice changes, facial and body hair growth as indicated in the scoring above) | Girls (add points relative to body hair and breast growth and consider menarche as indicated in the scoring above) |
|---|---|---|
| 1 (pre-pubertal) | 3 points | 3 points |
| 2 (early-pubertal) | 4–5 points (with no 3-point answers) | 3 points (with no menarche) |
| 3 (mid-pubertal) | 6–8 points (with no 4-point answers) | 4–8 points (with no menarche) |
| 4 (late-pubertal) | 9–11 points | 1–7 points (with menarche) |
| 5 (post-pubertal) | 12 points | 8 points (with menarche) |

*: For all questions except menarchal status: "not yet started" = 1 point, "barely started" = 2 points, "definitely started" = 3 points, "seems complete" = 4 points, "I don't know" = 0. For menarche, "no" = 1 point, "yes" = 4 points.

clearance at the relevant institutional or national research ethics board, the research team will seek the approval of the Ministry of Education (or similar national authorities) to visit local schools for subjects' recruitment. Once this approval is obtained, local schools will be approached and administrative approval to recruit students will be sought from the school directorate. All secondary school students, i.e. students in grades 10–12, will receive written information about the study, and copies of the parental consent form and adolescents assent forms (for students aged 15–17.9 years old). Those who agree to allow their child to participate will then return the signed parental consent form to the school. Moreover, the adolescents' assent form signed by the participating subject, should be returned to the school. Following a specific protocol, eligibility of adolescents who agreed to be part of the study, and whose parents/legal guardians have consented, will be confirmed based on a screening questionnaire that inquires about age, nationality, health/medication status, puberty stage, measured weight and height (and body mass index consequently). For participants aged 18 years old, their consent will be sought, without the need for the consent of parents/legal guardians, after which the same screening protocol will be applied.

In each country, eligible participants will be invited to visit the collaborative institution where the data collection process will be initiated. In an interview-setting, trained researchers will administer a multi-component questionnaire, inquiring about demographic and socioeconomic characteristics (age, sex, parental education level) as well as crowding index. Crowding index, a socio-economic proxy, is calculated as the ratio of the number of people living in the household over the number of rooms in the house used for sleeping [37].

The questionnaire will also inquire about lifestyle characteristics, including smoking, dietary behavior and physical activity, based on the Global School-based Student Health Survey (GSHS) [38]. Physical activity will be assessed by inquiring about the following: 1) number of days of being physically active for a total of at least 60 minutes/day during the past 7 days; 2) number of days of attending physical education class during the school year; and 3) time spent sitting and watching television, playing computer games, talking with friends or doing other sitting activities on a typical or usual day [38]. Dietary behavior will be assessed by the daily consumption of fruits, vegetables and carbonated soft drinks during the past 7 days, as well as fast foods intake during the past 7 days [38]. In addition, a one-day 24 hour dietary recall will be administered to the study subjects, using the multi-pass approach by trained nutritionists.

1. **Anthropometrics.** Anthropometric measurements will be obtained from all study participants, including weight, height, WC and MUAC. For weight measurement, participants will be instructed to be in light indoor clothing, barefoot or wearing stockings. The weight will be measured using a standard calibrated balance (Seca 869 Digital Floor Scale, Seca, Hamburg, Germany). Height will be measured using a wall-mounted Seca stadiometer (Seca 213, Seca, Hamburg, Germany), and without shoes. Weight and height are to be measured to the nearest 0.1 kg and 0.1 cm, respectively. The measurements will be taken twice and repeated a third time if the first two measurements differed by more than 0.3 kg (weight) or 0.5 cm (height). BMI will be calculated as weight (in kilogram) divided by the square of height (in meter) ($kg/m^2$). BMI z-scores for the subjects will be derived using the WHO Anthroplus software [39].

WC will be measured to the nearest 0.5 cm, using a non-stretching measuring tape (Seca 201, Seca, Hamburg, Germany) at the midpoint between the bottom of the rib cage and above the top of the iliac crest while the participants stand and follow normal expiration [21]. The measurement will be taken twice and the average of both values will be used. WC percentile will be determined based on the distribution published by Fernandez et al (2004) [40], given

the lack of local WC distribution charts in the region. WHtR will be obtained by dividing the WC (in cm) over height (in cm).

MUAC will be measured using a non-stretching measuring tape (Seca 201, Seca, Hamburg, Germany) and by taking half the distance between the acromion process at the back of the shoulder and the olecranon at the elbow level. It is measured to the nearest 1 mm.

2. **Body composition assessment**. Body composition will be assessed using the DDL technique as well as BIVA, and the proportions of BF and FFM will be determined.

**Deuterium dilution technique.**  The DDL technique will be used to assess TBW. Following Standard Operating Procedures, participants will be administered an oral dose of approximately 0.1 g $D_2O$/kg body weight (Cortecnet, Voisins-le-Bretonneux, France), after a 10% dilution in water. A saliva sample will be obtained prior to the dose and at 3 hours post-dose when the deuterium has reached equilibrium with the total body water, using a cotton swab. The pre-dosing saliva sample is used to correct the background $^2H_2O$ concentration values for the post-dose and all assays are performed in triplicates. Physical activity will not be allowed during the study day. As per the IAEA recommended procedure for deuterium dilution [41], all participants are given a standardized dry snack and a juice of approximately 200 ml about one hour after dose administration, and at least 30 minutes before the post-dose saliva sampling.

The deuterium enrichment in post-dose saliva is determined by Fourier Transform Infrared (FTIR) spectrometry (portable Agilent FTIR 4500s, Agilent, Santa Clara, United States). The dilution space is derived from the post-dose enrichment, and converted to TBW after adjustment for non-aqueous exchange of hydrogen atoms in the body (TBW = Dilution Space/1.04) [42]. FFM will be derived from TBW using a hydration coefficient, that is, the fraction of FFM comprised of water. Age- and gender-specific constants for hydration of FFM for adolescents will be used to calculate FFM [43]. The absolute FM will be derived by subtracting FFM from body weight, based on the two-compartment body composition model and %BF will then calculated. FM index (FMI) and FFMI are calculated as FM or FFM divided by height (m) squared.

**Bioelectrical impedance analysis (BIA).**  Body composition will also be assessed using the BIVA device (BIA 101, Akern, Via Lisbona, Italy) [44]. The device can measure TBW, tissue and water compartments, FM and FFM. BIA measurement will be taken on the right side of the body using electrodes and standardized testing procedures. Body composition values will be derived using the device's software without modification. Factors known to affect hydration status and BIVA measurement, such as room temperature, exercise, eating and drinking will be controlled. Before each testing session, the instrument will be checked using the test cell provided by the manufacturer. Calf circumference (CC) will also be measured (to the nearest 0.1 cm) by sliding the measuring tape (Seca 201, Seca, Hamburg, Germany) up and down to find the widest area on the right calf while the person remains seated. CC values will be used in BIVA calculations alongside MUAC and WC.

3. **Biochemical assessment and blood pressure measurement.** For the biochemical assessment, subjects will provide an 8-hour fasting blood sample. Prior to blood withdrawal, fingerprick glucose will be assessed to make sure the participants are fasting. Blood withdrawal will then be performed by certified phlebotomists or nurses. The samples will be centrifuged and stored immediately at -20 ˚C until the time of analysis. Samples will be analysed for fasting blood lipids including Triglycerides (TG), high-density lipoprotein cholesterol (HDL-C), low-density lipoprotein cholesterol (LDL-C), and total cholesterol

(TC), as well as glycemia (fasting glucose levels) at local certified laboratories, using standardized methodologies.

BP measurements will be obtained using a standard mercury sphygmomanometer (Omron 7 Series Upper Arm Blood Pressure machine monitor BP760); after participants are seated and rested for at least 5 minutes. The measurement will be taken twice and the average of both values will be used.

**Definition of the MetS.** MetS will be defined based on the harmonized definition [45], whereby participants will be classified as having the MetS if they have 3 out of the 5 following cardiometabolic abnormalities based on the International Diabetes Federation (IDF) cutoffs for subjects aged 16 years and above [46]: (i) elevated TG level ($\geq$150 mg/dl); (ii) low HDL-C level ($<$40 mg/dl for boys and $<$50 mg/dl for girls); (iii) elevated BP (systolic BP $\geq$130 mm Hg and/or diastolic BP $\geq$ 85 mm Hg); (iv) elevated fasting glucose level ($\geq$100 mg/dl); and (v) elevated WC ($\geq$94 cm for men, $\geq$80 cm for women). For subjects aged $<$ 16 years, the IDF recommends using the same cut-offs with the exception of WC and HDL. The cut-offs to be considered instead are as follows: WC $\geq$ 90th percentile for age and sex, HDL $<$ 40 mg/dl [46].

**Safety of the DDL technique.** Stable isotopes, including deuterium, are naturally found in the human body in small proportions referred to as the "natural abundance" [27]. The human body water contains around 0.015% of deuterium [27]. Stable isotopes have been extensively used in human studies for over half a century [27,47,48]. Deuterium, being a non-radioactive and non-toxic isotope, is easily eliminated from the body through urine, saliva and sweat once administered orally and after getting in contact with the body water [27,48]. The technique is completely safe and has been widely used in individuals of all ages, including infants and pregnant and lactating women [49]. It has been used in several studies conducted among children and adolescents in Spain [50], the United Kingdom (UK) [51] and New Zealand [52], as well as Asian countries such as China, Malaysia, Philippines, Thailand and Russia [53,54].

Transitory adverse effects, such as vertigo and nausea, have been reported upon consumption of deuterium at a level that causes more than 0.2% enrichment of body water [27,47]. The enrichment of deuterium oxide in human studies, however, is approximately 0.02% [48]. The threshold of deuterium toxicity in humans, defined at 15% labelled deuterium, is much higher than what is used in human studies [47,48]. Doses below this threshold have not been associated with any potentially harmful effects [27,47]. This has been acknowledged since 1979, when WA Coward ascertained, in a letter in the Lancet, that using deuterium in young infants was not a safety concern [55], reiterating that a single dose of deuterium oxide of 0.1 g/kg body weight (the dose that is used in the present study) would increase the deuterium concentration from a normal value of 150 p.p.m to 293 p.p.m, which is equivalent to 0.0293% of TBW, which is about 500 times less than the concentration required for toxic side effects [48].

**Ethical considerations.** Ethical approval for the study was already obtained from the Institutional Review Board of the American University of Beirut (BIO-2021-0022) in Lebanon, the Institutional Review Board of King Abdullah Bin Abdulaziz University Hospital (Log number 22–0029) in KSA, the Institutional Committee of Bioethics in Syria (date September 1, 2021), the Ministry of Health for Planning and Quality Affairs (Research number 2134/2022) in Kuwait, and the Ministry of Health (date June 3, 2021 –Number 23221 /8/1/2) in Jordan. Written informed assents from adolescents and written informed consents from parents or guardians will be sought. The study will be conducted according to the principles of the Declaration of Helsinki.

## Statistical analysis

Statistical analysis will be performed using the Statistical Analysis Package for Social Sciences (SPSS). Descriptive statistics will be presented to summarize the study variables of interest as counts and percentages for the categorical variables and as means and standard deviations (SD) for the continuous ones. A comparison between participants with and without MetS will be conducted using Chi-square for categorical variables and independent t-tests for continuous variables. Receiver operating characteristic (ROC) analyses will be performed to determine optimal cut-off values of BMI, WC, WHtR, %BF and MUAC in identifying those with MetS. The optimal cut-off point in each case will be estimated as the maximum value of Youden's index [sensitivity + specificity − 1]. Specific combinations of anthropometric indicators in predicting the MetS will also be investigated.

The validity of BIVA in body composition assessment will also be investigated. Differences in %BF estimates derived from DDL and BIVA will be assessed using paired t-test. Pearson correlation coefficients will be assessed to determine the correlation between %BF estimated by DDL and BIVA. The Bland and Altman approach [56] will also be adopted to compare BIVA against the reference DDL method for the determination of body composition parameters (%BF, TBW, body fat (kg) and FMI). Mean difference, 95%CI and limits of agreement (mean difference±2SD) between body composition parameters estimated via BIVA and DDL will be determined. A paired student's t test will be carried out to assess whether mean difference between DDL and BIVA is significantly different from zero. The closest the mean difference to zero, the higher the agreement between the reference method and the other method under investigation [35].

**Data management plan.** The International Atomic Energy Agency (IAEA)'s International Research Integration System (IRIS) will be used for data management. IRIS is an online direct data capture platform that allows to collect data of various structure and complexity with a data export and basic analysis possibility. Hence, all participating countries will be using the same data entry system. IRIS can be used as direct data entry via a browser if internet is accessible when collecting data, or it can be used to enter data from paper records. Data entry and access is password protected and storage is cloud-based. Data will be de-identified before analysis.

**Status and timeline of the study.**

| Activity | May-October 2023 | November 2023-April 2024 | May-October 2024 | November 2024-April 2025 |
|---|---|---|---|---|
| **Recruitment and data collection** | X[a] | X | X | |
| **Sample analysis (blood and saliva)[b]** | | X | X | X |
| **Data analysis and reporting** | | | X | X |

[a] Mainly recruitment and pilot-testing.

[b] Sample analysis pertinent to blood and saliva is planned to be initiated in parallel to data collection (in batches), but it is also expected to continue for a couple of months after data collection ends.

## Discussion

This study will be the first from the Middle-East to use a high quality reference method of body composition for determining ethnic-specific BF and anthropometric cut-offs among adolescents from the Middle-East, where a sharp increase in obesity prevalence and associated co-morbidities have been reported [57]. The study will also generate information on the validity of BIVA in assessing body composition among Middle-Eastern adolescents, against DDL as

the reference method. Findings stemming from this study will improve the screening for and diagnosis of excess adiposity among adolescents in the Middle-East, improve public health surveillance of excess adiposity, and contribute to better prevention and management strategies.

This study tackles a public health priority that is common to all countries of the Middle-East, where the nutrition transition is unfolding, with its characteristic shifts in diet, lifestyle and body composition [3]. It has been proposed that these shifts may be more exaggerated in adolescents, given their higher exposure to the "modern" food environment, including food advertisements and promotion. Moreover, adolescents tend to engage in unhealthy eating practices such as eating outside the home and consuming highly processed energy-dense foods [58–60]. This may at least partially explain the sky-rocketing levels of adolescent obesity reported from Middle-Eastern countries [57], increasing the risk for NCDs and raising the associated health care cost in countries of the region. Acknowledging that adolescents represent approximately 18% of the population of the Middle-East, and that adolescent obesity is likely to track into adulthood, the need for developing population-specific strategies aimed at improving the diagnosis of excess adiposity and developing tailored prevention and management interventions, ought to be highlighted. It is worth noting that the prevalence of obesity and related metabolic abnormalities may have also increased in the region during the prolonged Covid-19 lockdowns of 2020–2021 [61], further highlighting the need for proper assessment and management of adiposity in this age group. This study therefore provides a unique opportunity to examine the body composition of adolescents in various countries of the region, and determine ethnic- and sex-specific cut-offs that may be adopted by policy makers in the region. The project will also be a platform for sharing knowledge among Middle-Eastern countries, fostering the optimal use of available resources.

In this study, ethnic-specific anthropometric cutoffs for the identification of the MetS, a constellation of abnormalities that are associated with obesity will be determined [9–11]. The selected anthropometric indicators include BMI Z-score, and WC percentile, which are commonly used in the screening of obesity, in addition to WHtR [13–15], which accounts for height in addition to WC, and was suggested as an additional indicator of abdominal adiposity and increased cardiovascular risk [62–66]. MUAC, which was traditionally an indicator of underweight, will also be measured in this study as a potential simple and practical indicator for the identification of high body fatness in children and adolescents [26]. In addition, and since the pathology and morbidity associated with obesity is primarily driven by excess fat mass, the screening for and monitoring of this condition should also directly assess adiposity. In this study, %BF will be determined using the DDL technique, which is a valid technique for body composition in different age groups [67–69]. In a study on Portuguese adolescents aged 10–18 years old, Quiterio et al. [70] showed that the DDL technique is an accurate and precise gold standard technique for body composition assessment. Acknowledging that DDL may not be suitable for routine screening and assessment, this study will also validate the use of other BF assessment methods that are more applicable in practice or field studies, including BIVA [28,29]. In this validation aspect, the DDL method will be considered the reference method.

The strengths of this multi-country study include the collection of anthropometric measurements and biochemical data according to standardized protocols and equipment, which contribute towards the reduction of measurement errors. Another strength is the recruitment of study participants from schools rather than enrolling those who were seeking health check-ups in medical centers, and who therefore may have higher levels of health consciousness [62]. However, the study design may be limited by several considerations, such as the cross-sectional design, which cannot be used to establish causality or temporal relationships [62]. In addition, the identification of metabolic abnormalities in this study is restricted to those included in the IDF definition of the MetS and thus other parameters that may be influenced

by obesity such as liver function tests or inflammatory markers are excluded. Moreover, in the absence of a validated food frequency questionnaire for countries in the region, dietary assessment is conducted based on a 24 hour dietary recall. Finally, this study does not comprise a comprehensive set of anthropometric indicators but rather focuses on the most common and field-appropriate ones, including BMI, WC, WHtR and MUAC, as these were shown to be practical, inexpensive and relatively easy to use in large field surveys" [71–77].

In conclusion, acknowledging that excess BF rather than excess body mass is related to increased disease risk, this project responds to the need for the identification of BF cut-off values in adolescents and hence, addresses an important knowledge gap in the literature. In addition, this project responds to the need for ethnic-specific anthropometric cut-offs for the identification of excess adiposity and associated cardiometabolic risks in the adolescent population of the Middle-East. It will specifically allow for a better understanding of the relationship between BF and simple anthropometry in this age group, and contribute to the derivation of optimal cut-off values of BMI, WC, WHtR, %BF and MUAC. The adoption of these population-specific cut-offs may support policy makers, public health experts and healthcare providers in developing ethnic-specific preventive and curative interventions that are better tailored to adolescents in the region. Future studies may examine the validity of the generated cut-offs in other countries of the region.

## Author Contributions

**Conceptualization:** Lara Nasreddine, Mohamad Adel Bakir, Pernille Kaestel, John J. Reilly.

**Data curation:** Lara Nasreddine, Mohamad Adel Bakir, Tareq Al-Ati, Abeer Salman Alzaben, Rawhieh Barham, Nahla Bawazeer, Elie-Jacques Fares, Kholoud Hammad, Mandy Taktouk.

**Methodology:** Lara Nasreddine, Pernille Kaestel, John J. Reilly.

**Writing – original draft:** Lara Nasreddine, Pernille Kaestel, John J. Reilly.

**Writing – review & editing:** Lara Nasreddine, Mohamad Adel Bakir, Tareq Al-Ati, Abeer Salman Alzaben, Rawhieh Barham, Nahla Bawazeer, Elie-Jacques Fares, Kholoud Hammad, Pernille Kaestel, John J. Reilly, Mandy Taktouk.

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
