## [Decision Letter · Decision Letter 0]

4 Jun 2023

PONE-D-23-02201Nutrition and Health in Arab Adolescents (NaHAR): Study Protocol for the Determination of Ethnic-Specific Body Fat and Anthropometric Cut-offs to Identify Metabolic SyndromePLOS ONE

Dear Dr. Nasreddine,

Thank you for submitting your manuscript to PLOS ONE. After careful consideration, we feel that it has merit but does not fully meet PLOS ONE’s publication criteria as it currently stands. Therefore, we invite you to submit a revised version of the manuscript that addresses the points raised during the review process.

We look forward to receiving your revised manuscript.

Kind regards,

Inge Roggen, M.D., Ph.D.

Academic Editor

PLOS ONE

Journal Requirements:

https://www.liebertpub.com/doi/10.1089/met.2019.0033

https://www.cambridge.org/core/journals/british-journal-of-nutrition/article/erosion-of-the-mediterranean-diet-among-adolescents-evidence-from-an-eastern-mediterranean-country/569D34E53A9274EF4A37AE0DD6C81113

In your revision ensure you cite all your sources (including your own works), and quote or rephrase any duplicated text outside the methods section. Further consideration is dependent on these concerns being addressed.

“The study is funded by International Atomic Energy Agency (IAEA), and supported through collaboration among ARASIA State Parties under the framework of the IAEA technical cooperation project RAS6094, ‘Applying Nuclear Techniques for the Determination of Body Fat and Anthropometric Cutoffs (ARASIA)’. (ARASIA is a Cooperative Agreement for Arab States in Asia for Research, Development and Training related to Nuclear Science and Technology).

There are no grants awarded to individual researchers in the respective countries. Instead, each of the countries receive  the supplies and deuterium from the IAEA but there are no grants per se.”

Reviewers' comments:

Reviewer's Responses to Questions

**Comments to the Author**

1. Does the manuscript provide a valid rationale for the proposed study, with clearly identified and justified research questions?

Reviewer #1: Yes

Reviewer #2: Yes

2. Is the protocol technically sound and planned in a manner that will lead to a meaningful outcome and allow testing the stated hypotheses?

Reviewer #1: Yes

Reviewer #2: Partly

3. Is the methodology feasible and described in sufficient detail to allow the work to be replicable?

Reviewer #1: Yes

Reviewer #2: Yes

4. Have the authors described where all data underlying the findings will be made available when the study is complete?

Reviewer #1: No

Reviewer #2: No

5. Is the manuscript presented in an intelligible fashion and written in standard English?

Reviewer #1: Yes

Reviewer #2: Yes

6. Review Comments to the Author

You may also provide optional suggestions and comments to authors that they might find helpful in planning their study.

Reviewer #1: Journal: PLOS ONE

Title: Nutrition and Health in Arab Adolescents (NaHAR): Study Protocol for the

Determination of Ethnic-Specific Body Fat and Anthropometric Cut-offs to Identify

Metabolic Syndrome

Reviewer’s comments

Sample size:

The manuscript used 210 and a minimum of 210 interchangeably for sample size. The authors should be clear about how many subjects will be included in each country.

Under data collection:

How many schools will be included in each country needs to be made clear. If your sample size is 210 and all students in grades 10-12 are invited to participate, what do you do with those children whose parents have consented for them to participate after you reach your sample size? It is unclear how the sample size will be maintained unless you have a pre-determined number of schools selected for each country and the number of students to participate in the study per school. Extra care should also be taken not to overburden parents when you know that only a few students will be selected for the study in a particular school.

Under Anthropometrics:

Provide the company(manufacturer) names and locations for each tool used. E.g., Seca scale, measuring tape for WC and MUAC, D2O for DDL, and Fourier Transform Infrared (FTIR) spectrometry. Provide details similar to what you did for the BIVA device (BIA 101, Akern, Via Lisbona, Italy)

WHO Anthroplus software(provide a citation)

Ethical considerations:

“Country-specific data will be pooled within a regional database for data analysis.” it is unclear why this statement is here.

Statistical analysis:

Lines#309,310: Chi-square and independent t-tests. Specify which test is for categorical and which is for continuous variables.

Lines#317,318 and 322-324. What is the purpose of performing repeated ANOVA and paired t-tests on DDL vs. BIVA? What additional information will both tests provide instead of just one?

Data management plan:

Fix the typo in the second sentence.

Line# 335. Use sentence case formatting as done for the other headings.

Reviewer #2: The manuscript entitled "Nutrition and Health in Arab Adolescents ( 1 NaHAR): Study Protocol for the Determination of Ethnic-Specific Body Fat and Anthropometric Cut-offs to Identify Metabolic Syndrome” is led by Nasreddine L et al., from a multicenter and multinational group of investigators. The strong team is set to achieve the goals listed in the protocol review. The study will contribute to the validation of anthropometric cut-off points in the middle-east population of adolescents with excess adiposity and associated metabolic syndrome risks.

The following points are written to improve their study:

1. Investigators from Jordan is still not have approved ethical clearance. Authors should clearly state whether it is okay to have this protocol publication before the approval of investigation in one site.

2. How was the sample size calculated? The sample size of 210 is sufficient for their population, without statistical analysis for sample size, this becomes a weakness.

3. Authors should consider food frequency questionnaire (FFQ) to monitor dietary intake

4. Authors can also consider measuring liver function test in the blood, as part of metabolic syndrome and this data will be useful in addition not lipid biomarkers in the adolescent population.

7. PLOS authors have the option to publish the peer review history of their article (what does this mean?). If published, this will include your full peer review and any attached files.

Reviewer #1: No

Reviewer #2: No

---

## [Author Response · Author response to Decision Letter 0]

12 Jul 2023

Journal Requirements:

Response: The manuscript has been reviewed and edited based on the PLOS ONE’s style templates provided by the editor.

https://www.liebertpub.com/doi/10.1089/met.2019.0033

https://www.cambridge.org/core/journals/british-journal-of-nutrition/article/erosion-of-the-mediterranean-diet-among-adolescents-evidence-from-an-eastern-mediterranean-country/569D34E53A9274EF4A37AE0DD6C81113

In your revision ensure you cite all your sources (including your own works), and quote or rephrase any duplicated text outside the methods section. Further consideration is dependent on these concerns being addressed.

Response: Thank you for your comment. The sections have been revised. The above mentioned references were also cited.

Response: Sure, will do.

“The study is funded by International Atomic Energy Agency (IAEA), and supported through collaboration among ARASIA State Parties under the framework of the IAEA technical cooperation project RAS6094, ‘Applying Nuclear Techniques for the Determination of Body Fat and Anthropometric Cutoffs (ARASIA)’. (ARASIA is a Cooperative Agreement for Arab States in Asia for Research, Development and Training related to Nuclear Science and Technology).

There are no grants awarded to individual researchers in the respective countries. Instead, each of the countries receive the supplies and deuterium from the IAEA but there are no grants per se.”

Response: The role of the funder has been clarified, as follows: 

“This work was supported by the International Atomic Energy Agency (Technical Cooperation Programme; RAS6094). The IAEA provided technical support to the countries and participated in the design, preparation, review, and approval of the manuscript.”

Response: This has been included in the cover letter.

Response: The reference list has been reviewed. We ensure that it is complete, correct and without any retracted articles.

Reviewers' comments:

Reviewer's Responses to Questions

Comments to the Author

1. Does the manuscript provide a valid rationale for the proposed study, with clearly identified and justified research questions?

Reviewer #1: Yes

Reviewer #2: Yes

2. Is the protocol technically sound and planned in a manner that will lead to a meaningful outcome and allow testing the stated hypotheses?

Reviewer #1: Yes

Reviewer #2: Partly

3. Is the methodology feasible and described in sufficient detail to allow the work to be replicable?

Reviewer #1: Yes

Reviewer #2: Yes

4. Have the authors described where all data underlying the findings will be made available when the study is complete?

Reviewer #1: No

Reviewer #2: No 

Response: Data underlying the findings will be provided through a supporting file or a database link, when the study is complete.

5. Is the manuscript presented in an intelligible fashion and written in standard English?

Reviewer #1: Yes

Reviewer #2: Yes

6. Review Comments to the Author

You may also provide optional suggestions and comments to authors that they might find helpful in planning their study.

Reviewer #1: Journal: PLOS ONE

Title: Nutrition and Health in Arab Adolescents (NaHAR): Study Protocol for the

Determination of Ethnic-Specific Body Fat and Anthropometric Cut-offs to Identify

Metabolic Syndrome

Reviewer’s comments

Sample size:

The manuscript used 210 and a minimum of 210 interchangeably for sample size. The authors should be clear about how many subjects will be included in each country.

Response: Thank you for pointing this to us. We have removed the term “minimum” as the sample size from each country is 210 adolescents. This has been clarified in the revised manuscript.

Under data collection:

How many schools will be included in each country needs to be made clear. If your sample size is 210 and all students in grades 10-12 are invited to participate, what do you do with those children whose parents have consented for them to participate after you reach your sample size? It is unclear how the sample size will be maintained unless you have a pre-determined number of schools selected for each country and the number of students to participate in the study per school. Extra care should also be taken not to overburden parents when you know that only a few students will be selected for the study in a particular school.

Response: We thank the reviewer for this comment. We have clarified in our revised manuscript that 4-5 schools will be approached in each country, whereby 45-50 students will be recruited from each school. This has been stated in the manuscript as follows: “In each country, a sample of 210 adolescents aged 15-18 years will be recruited from 4-5 local schools.” (Lines 142-143).

As for the consent process and based on previous studies, refusal rate is usually high in such studies involving adolescents. We will make sure to keep a daily log on the number of participants we have recruited before inviting others to join.

Under Anthropometrics:

Provide the company(manufacturer) names and locations for each tool used. E.g., Seca scale, measuring tape for WC and MUAC, D2O for DDL, and Fourier Transform Infrared (FTIR) spectrometry. Provide details similar to what you did for the BIVA device (BIA 101, Akern, Via Lisbona, Italy)

WHO Anthroplus software(provide a citation)

Response: The company names and locations have been added for the seca scale, stadiometer, measuring tape for WC, MUAC and calf circumference, as well as the D2O and FTIR. Moreover, we provided the following citation for the Anthroplus software:

World Health Organization. Growth reference data for 5-19 years. Application tools: WHO AnthroPlus software. [cited 2023 July 4]. Available from: https://www.who.int/tools/growth-reference-data-for-5to19-years/application-tools.

Ethical considerations:

“Country-specific data will be pooled within a regional database for data analysis.” it is unclear why this statement is here. 

Response: We thank you for pointing this to us. The statement has been removed from the manuscript.

Statistical analysis:

Lines#309,310: Chi-square and independent t-tests. Specify which test is for categorical and which is for continuous variables. 

Response: Based on the reviewer’s comment, we have revised this section, as follows: 

“A comparison between participants with and without MetS will be conducted using Chi-square for categorical variables and independent t-tests for continuous variables”. (Lines 321-323)

Lines#317,318 and 322-324. What is the purpose of performing repeated ANOVA and paired t-tests on DDL vs. BIVA? What additional information will both tests provide instead of just one? 

Response: We thank you for pointing this to us. There is in fact no need for ANOVA. We have edited this section, as follows: “Differences in %BF estimates derived from DDL and BIVA will be assessed using paired t-test”. (Line 330)

In fact, this analysis will help us in addressing our secondary objective related to examining the validity of Bioelectrical Impendence Vector Analysis (BIVA) in estimating BF against the deuterium dilution technique (DDL). 

Line# 335. Use sentence case formatting as done for the other headings.

Response: Done.

Reviewer #2: The manuscript entitled "Nutrition and Health in Arab Adolescents ( 1 NaHAR): Study Protocol for the Determination of Ethnic-Specific Body Fat and Anthropometric Cut-offs to Identify Metabolic Syndrome” is led by Nasreddine L et al., from a multicenter and multinational group of investigators. The strong team is set to achieve the goals listed in the protocol review. The study will contribute to the validation of anthropometric cut-off points in the middle-east population of adolescents with excess adiposity and associated metabolic syndrome risks.

The following points are written to improve their study:

1. Investigators from Jordan is still not have approved ethical clearance. Authors should clearly state whether it is okay to have this protocol publication before the approval of investigation in one site. 

Response: Jordan has obtained ethical clearance. The statement has been edited as follows: “Ethical approval for the study was already obtained from the Institutional Review Board of the American University of Beirut (BIO-2021-0022) in Lebanon, the Institutional Review Board of King Abdullah Bin Abdulaziz University Hospital (Log number 22-0029) in KSA, the Institutional Committee of Bioethics in Syria (date September 1, 2021), the Ministry of Health for Planning and Quality Affairs (Research number 2134/2022) in Kuwait, and the Ministry of Health (date June 3, 2021 – Number 23221 /8/1/2) in Jordan.” (Lines 306-311)

2. How was the sample size calculated? The sample size of 210 is sufficient for their population, without statistical analysis for sample size, this becomes a weakness.

Response: The sample size was calculated using the STEP sample size calculator. This has been clarified in the manuscript as follows: “The sample size of 210 subjects was calculated using the STEPS Sample Size Calculator and Sampling Spreadsheet, using a margin of error of 5%, a confidence interval (CI) of 95% and a prevalence of MetS among adolescents of 10%.” (Lines 148-151)

Reference: World Health Organization. STEPS Sample Size Calculator and Sampling Spreadsheet. [cited 2020 Nov 9]. Available from: https://cdn.who.int/media/docs/default-source/ncds/ncd-surveillance/steps/sample-size-calculator.xls?sfvrsn=ee1f4ae8_2.

3. Authors should consider food frequency questionnaire (FFQ) to monitor dietary intake 

Response: We thank you for this suggestion. Unfortunately, we do not have a validated FFQ in each of the countries where data collection is being conducted.

4. Authors can also consider measuring liver function test in the blood, as part of metabolic syndrome and this data will be useful in addition not lipid biomarkers in the adolescent population. 

Response: Indeed this would provide valuable insight as liver enzymes may be affected by obesity and metabolic abnormalities. However, we have opted to adhere to the official definition of the metabolic syndrome. For consistency purposes, we have also removed glycated Hb measurements from the manuscript, so we are fully in line with the official definition of the MetS. 

We have also addressed this point in the limitations section, as follows:

“In addition, the identification of metabolic abnormalities in this study is restricted to those included in the IDF definition of the MetS and thus other parameters that may be influenced by obesity such as liver function tests or inflammatory markers are excluded.”

7. PLOS authors have the option to publish the peer review history of their article (what does this mean?). If published, this will include your full peer review and any attached files.

Do you want your identity to be public for this peer review? For information about this choice, including consent withdrawal, please see our Privacy Policy.

Reviewer #1: No

Reviewer #2: No

---

## [Decision Letter · Decision Letter 1]

4 Sep 2023

PONE-D-23-02201R1Nutrition and Health in Arab Adolescents (NaHAR): Study Protocol for the Determination of Ethnic-Specific Body Fat and Anthropometric Cut-offs to Identify Metabolic SyndromePLOS ONE

Dear Dr. Nasreddine,

Thank you for submitting your manuscript to PLOS ONE. After careful consideration, we feel that it has merit but does not fully meet PLOS ONE’s publication criteria as it currently stands. Therefore, we invite you to submit a revised version of the manuscript that addresses the points raised during the review process.

 This manuscript requires a minor revisionThe authors should sufficiently address the minor concerns raised by the reviewer Specific feedbacks are provided by reviewers below==============================

We look forward to receiving your revised manuscript.

Kind regards,

Fredirick Lazaro mashili, MD, PhD

Academic Editor

PLOS ONE

Journal Requirements:

Additional Editor Comments:

The authors should address the minor comment given by the reviewer

Reviewers' comments:

Reviewer's Responses to Questions

**Comments to the Author**

1. Does the manuscript provide a valid rationale for the proposed study, with clearly identified and justified research questions?

Reviewer #2: Yes

Reviewer #3: Yes

2. Is the protocol technically sound and planned in a manner that will lead to a meaningful outcome and allow testing the stated hypotheses?

Reviewer #2: Yes

Reviewer #3: Yes

3. Is the methodology feasible and described in sufficient detail to allow the work to be replicable?

Reviewer #2: Yes

Reviewer #3: Yes

4. Have the authors described where all data underlying the findings will be made available when the study is complete?

Reviewer #2: No

Reviewer #3: Yes

5. Is the manuscript presented in an intelligible fashion and written in standard English?

Reviewer #2: Yes

Reviewer #3: Yes

6. Review Comments to the Author

You may also provide optional suggestions and comments to authors that they might find helpful in planning their study.

Reviewer #2: Authors have improved their protocol. I would still suggest them to include Harvard Willet FFQ for their study and liver function test would be additional work, but would provide great knowledge. However, this is a protocol and their choice is okay

Reviewer #3: INTRODUCTION

Lines 112 and 113

•Why did the authors not put under consideration other core elements for anthropometry such as waist-to-hip ratio, thigh-to-hip ratio, and skinfold thickness? This has also been used in measuring adiposity.

Line 121- 124.

•I would advise revising the sentence to be more inclusive of the study objectives, not only anthropometrics cut-offs. i.e. will the study not bring validated Body Fats measurements?

MATERIALS AND METHODS

Table 1. What types of illnesses among the wide range of inborn metabolic disorders do you specifically mean when you say "those with inborn errors of metabolism"? and How will they be recognized for the purpose of exclusion

Deuterium Dilution Technique:

•Due to the circadian effect on saliva. It is recommended to collect saliva at a specific time. Does this apply to this study? Authors should revise thoroughly on methods for saliva collection. Including the exclusion of participants with any oral illness like xerostomia and others that may affect the saliva flow rate and therefore distort the results. For more insights about saliva collection methods authors can read this paper https://doi.org/10.3390/oral3030027

•Will participants be allowed to eat or drink anything before post-dose saliva collection?

•If “Yes” will it not bring errors in Salimetrics

•If "No," authors should explain how they will restrict eating since participants must fast for at least eight hours in

order to measure blood parameters that need fasting and as a prerequisite for saliva collection. Three more hours

would jeopardize the participants and some may hesitate to participate.

Status and Timeline of the study

The table is not clear. Have the authors begun the data collection?

7. PLOS authors have the option to publish the peer review history of their article (what does this mean?). If published, this will include your full peer review and any attached files.

Reviewer #2: No

Reviewer #3: **Yes: **Oscar Mbembela

---

## [Author Response · Author response to Decision Letter 1]

19 Oct 2023

October 16, 2023

Nutrition and Health in Arab Adolescents (NaHAR): Study Protocol for the Determination of Ethnic-Specific Body Fat and Anthropometric Cut-offs to Identify Metabolic Syndrome

Manuscript ID: PONE-D-23-02201R1

We would like to thank the editor and the reviewers for the constructive comments on our manuscript. Please find below a point by point response to each of the comments provided by the reviewers.

Journal Requirements:

Response to comment: The reference list has been reviewed to ensure that it is complete, correct and without any retracted articles.

Additional Editor Comments:

The authors should address the minor comment given by the reviewer

Reviewers' comments:

Reviewer's Responses to Questions

Comments to the Author

1. Does the manuscript provide a valid rationale for the proposed study, with clearly identified and justified research questions?

Reviewer #2: Yes

Reviewer #3: Yes

2. Is the protocol technically sound and planned in a manner that will lead to a meaningful outcome and allow testing the stated hypotheses?

Reviewer #2: Yes

Reviewer #3: Yes

3. Is the methodology feasible and described in sufficient detail to allow the work to be replicable?

Reviewer #2: Yes

Reviewer #3: Yes

4. Have the authors described where all data underlying the findings will be made available when the study is complete?

Reviewer #2: No

Reviewer #3: Yes

Response to comment: We have added the following statement to our protocol paper in order to clarify where the data underlying the findings can be found:

“Data underlying the findings will be provided through a supporting file or a database link, when the study is complete.” (Lines 423-424)

5. Is the manuscript presented in an intelligible fashion and written in standard English?

Reviewer #2: Yes

Reviewer #3: Yes

6. Review Comments to the Author

You may also provide optional suggestions and comments to authors that they might find helpful in planning their study.

Reviewer #2: 

Authors have improved their protocol. I would still suggest them to include Harvard Willet FFQ for their study and liver function test would be additional work, but would provide great knowledge. However, this is a protocol and their choice is okay

Response to comment: We thank the reviewer for the valuable comments on our manuscript. We agree that a food frequency questionnaire would provide important data on the habitual intake of participating subjects. Although the Harvard Willet FFQ has been shown to be a valid assessment tool in the USA (and other countries with similar dietary practices), its validity has not been ascertained in countries that may have significantly different dietary habits (such as countries of the region). There is an urgent need to develop and validate a FFQ that is specific to Arab countries. We have acknowledged this point in our limitations section, as follows: 

“Moreover, in the absence of a validated food frequency questionnaire for countries in the region, dietary assessment is conducted based on a 24 hour dietary recall.” (Lines 397-399)

We do also agree with the added value of liver tests, however it will be difficult for us to implement it because of budgetary implications as well as for consistency with the official definition of the metabolic syndrome. We have acknowledged this in the limitation section as follows: 

 “In addition, the identification of metabolic abnormalities in this study is restricted to those included in the IDF definition of the MetS and thus other parameters that may be influenced by obesity such as liver function tests or inflammatory markers are excluded.” (Lines 394-397)

Reviewer #3: 

INTRODUCTION

Lines 112 and 113

•Why did the authors not put under consideration other core elements for anthropometry such as waist-to-hip ratio, thigh-to-hip ratio, and skinfold thickness? This has also been used in measuring adiposity.

Response to comment: We thank the reviewer for this comment. Indeed these measurements/ratios (waist to hip, thigh to hip; skinfold thicknesses) have been previously used in assessing adiposity. However, in our study, we tried to focus on the most common, and field-appropriate anthropometric indicators including BMI, WC, WHtR and MUAC. These indicators are practical, inexpensive and relatively easy to use in large field surveys. 

Although useful in certain contexts, skinfold thicknesses are difficult to measure with precision and accuracy without rigorous training, and hence may not be very suited to field-based assessments. These measurements have been also described to be subject to considerable inter- and intra-observer error in the measurements. It may be difficult to pick up a consistent fold of skin and subcutaneous fat and, without proper care and attention, the use of the calipers may causes pain or bruising. In the very obese, the skinfold may also sometimes be larger than the calipers can measure.

References: 

Wendel, D., Weber, D., Leonard, M. B., Magge, S. N., Kelly, A., Stallings, V. A., ... & Zemel, B. S. (2017). Body composition estimation using skinfolds in children with and without health conditions affecting growth and body composition. Annals of human biology, 44(2), 108-120.

Eaton–Evans, J. (2005). Nutritional assessment| Anthropometry. https://www.sciencedirect.com/science/article/pii/B9780123750839001975?via%3Dihub

The waist to hip ratio may also provide valuable information on body fat distribution, but it will necessitate obtaining additional measurements of the hip circumferences which may decrease its practical acceptability or feasibility. In addition, various studies have suggested that waist circumference or waist to height ratio are better correlated with body fat and/or cardiovascular risk than the waist to hip ratio.

References: 

Ashtary-Larky, D., Daneghian, S., Alipour, M., Rafiei, H., Ghanavati, M., Mohammadpour, R., ... & Afrisham, R. (2018). Waist circumference to height ratio: better correlation with fat mass than other anthropometric indices during dietary weight loss in different rates. International journal of endocrinology and metabolism, 16(4).

Bacopoulou, F., Efthymiou, V., Landis, G., Rentoumis, A., & Chrousos, G. P. (2015). Waist circumference, waist-to-hip ratio and waist-to-height ratio reference percentiles for abdominal obesity among Greek adolescents. BMC pediatrics, 15(1), 1-9.

Gažarová, M., Bihari, M., Lorková, M., Lenártová, P., & Habánová, M. (2022). The Use of different anthropometric indices to assess the body composition of Young Women in Relation to the incidence of obesity, Sarcopenia and the premature mortality risk. International Journal of Environmental Research and Public Health, 19(19), 12449.

Caminha, T. C., Ferreira, H. S., Costa, N. S., Nakano, R. P., Carvalho, R. E. S., Xavier Jr, A. F., & Assunção, M. L. (2017). Waist-to-height ratio is the best anthropometric predictor of hypertension: a population-based study with women from a state of northeast of Brazil. Medicine, 96(2).

We agree that there is potential for thigh circumference or the thigh-to-hip ratio to be a proxy for adiposity (or LBM). However, the use of these indicators may be further complicated by its cultural acceptability and feasibility.

Reference:

McCarthy, H. D. (2014). Conference on ‘Childhood nutrition and obesity: current status and future challenges’ Symposium 2: Data collection; Measuring growth and obesity across childhood and adolescenc. PROCEEDINGS OF THE NUTRITION SOCIETY, 73, 210-217.

We have acknowledged this point in the limitations section as follows: “Finally, this study does not comprise a comprehensive set of anthropometric indicators but rather focuses on the most common and field-appropriate ones, including BMI, WC, WHtR and MUAC, as these were shown to be practical, inexpensive and relatively easy to use in large field surveys.” (Lines 399-402). 

Line 121- 124.

•I would advise revising the sentence to be more inclusive of the study objectives, not only anthropometrics cut-offs. i.e. will the study not bring validated Body Fats measurements?

Response to comment: as recommended by the reviewer, this sentence has been revised to the following: “This study will therefore contribute to the generation of validated anthropometric and body fat cut-off points”.

MATERIALS AND METHODS

Table 1. What types of illnesses among the wide range of inborn metabolic disorders do you specifically mean when you say "those with inborn errors of metabolism"? and How will they be recognized for the purpose of exclusion.

Response to comment: We thank the reviewer for this comment. We have clarified in our revised manuscript that this exclusion will be based on whether the subject reports a previous diagnosis of inborn errors of intermediary metabolism (that are mainly related amino acids, carbohydrates, and fatty acids’ metabolism). This will be ascertained via a screening questionnaire that will be administered to the subjects to assess their eligibility to participate in the study.

To address the reviewer’s comment, Table 1 was modified as well as lines 171-175 where we clarified the following: “Following a specific protocol, eligibility of adolescents who agreed to be part of the study, and whose parents/legal guardians have consented, will be confirmed based on a screening questionnaire that inquires about age, nationality, health/medication status, puberty stage, measured weight and height (and body mass index consequently”. 

Deuterium Dilution Technique:

•Due to the circadian effect on saliva. It is recommended to collect saliva at a specific time. Does this apply to this study? Authors should revise thoroughly on methods for saliva collection. Including the exclusion of participants with any oral illness like xerostomia and others that may affect the saliva flow rate and therefore distort the results. For more insights about saliva collection methods authors can read this paper https://doi.org/10.3390/oral3030027

Response to comment: We thank you for your comment. 

As suggested by the reviewer, we have amended the exclusion criteria to exclude those reporting to have oral illnesses like xerostomia and others that may affect the saliva flow. This is shown in Table 1 in the revised manuscript. 

We agree that there may well be circadian variation in saliva production and in saliva composition, but not in relation to the behaviour of the isotope in the period between dosing and post dose sampling (Coward 1982). Since Deuterium is in equilibrium with Hydrogen throughout the body water after 3-4 hours, saliva production or composition is not considered to have an influence on Deuterium enrichment (Coward 1982)..

Reference: Coward, W. A., Cole, T. J., Sawyer, M. B., & Prentice, A. M. (1982). Breast-milk intake measurement in mixed-fed infants by administration of deuterium oxide to their mothers. Human nutrition. Clinical nutrition, 36(2), 141-148.

•Will participants be allowed to eat or drink anything before post-dose saliva collection? •If “Yes” will it not bring errors in Salimetrics

•If "No," authors should explain how they will restrict eating since participants must fast for at least eight hours in

order to measure blood parameters that need fasting and as a prerequisite for saliva collection. Three more hours would jeopardize the participants and some may hesitate to participate

Response to comment: Thank you for this comment. Yes, we have clarified this point in the revised manuscript as follows: “As per the IAEA recommended procedure for deuterium dilution, all participants are given a standardized dry snack and a juice of approximately 200 ml about one hour after dose administration, and at least 30 minutes before the post-dose saliva sampling.” (Lines 225-228). 

In fact this IAEA procedural guidance document has explained that “To allow full and fast absorption of the dose, it is recommended that the participant does not drink or eat anything until 30 minutes after dosing. In addition, to avoid dilution of the deuterium enrichment in saliva, no food or drink should be taken during the last 30 minutes before the post-dose saliva samples are collected. Consider giving a standardized snack and a drink (preferably water) of up to 250 mL to children and adults 1 to 2 hours after dosing.”. 

Status and Timeline of the study

The table is not clear. Have the authors begun the data collection?

Response to comment: The table has been amended as the timeline of the project has changed since our first submission to the journal. We have also provided additional clarifications as a legend to the table. 

Recruitment and data collection (mainly pilot-testing) was initiated in some sites. Sample analysis pertinent to blood and saliva is planned to be initiated in parallel to data collection (in batches), but it is also expected to continue for a couple of months after data collection ends (i.e. beginning of year 2025). Data analysis and reporting will be performed during 2024 and beginning of 2025.

7. PLOS authors have the option to publish the peer review history of their article (what does this mean?). If published, this will include your full peer review and any attached files.

Do you want your identity to be public for this peer review? For information about this choice, including consent withdrawal, please see our Privacy Policy.

Reviewer #2: No

Reviewer #3: Yes: Oscar Mbembela

---

## [Decision Letter · Decision Letter 2]

23 Jan 2024

Nutrition and Health in Arab Adolescents (NaHAR): Study Protocol for the Determination of Ethnic-Specific Body Fat and Anthropometric Cut-offs to Identify Metabolic Syndrome

PONE-D-23-02201R2

Dear Dr. Nasreddine,

We’re pleased to inform you that your manuscript has been judged scientifically suitable for publication and will be formally accepted for publication once it meets all outstanding technical requirements.

Kind regards,

Fredirick Lazaro mashili, MD, PhD

Academic Editor

PLOS ONE

Additional Editor Comments (optional):

All the comments raised by all the teviewers have been sufficiently addressed

Reviewers' comments:

Reviewer's Responses to Questions

**Comments to the Author**

1. Does the manuscript provide a valid rationale for the proposed study, with clearly identified and justified research questions?

Reviewer #3: Yes

Reviewer #4: Yes

2. Is the protocol technically sound and planned in a manner that will lead to a meaningful outcome and allow testing the stated hypotheses?

Reviewer #3: Yes

Reviewer #4: Yes

3. Is the methodology feasible and described in sufficient detail to allow the work to be replicable?

Reviewer #3: Yes

Reviewer #4: Yes

4. Have the authors described where all data underlying the findings will be made available when the study is complete?

Reviewer #3: Yes

Reviewer #4: Yes

5. Is the manuscript presented in an intelligible fashion and written in standard English?

Reviewer #3: Yes

Reviewer #4: Yes

6. Review Comments to the Author

You may also provide optional suggestions and comments to authors that they might find helpful in planning their study.

Reviewer #3: After reading the responses to the antecedent comments, I'm confident that they have been well-defended, thus this work can be approved for publication.

Reviewer #4: The authors have sufficiently addressed all the comments raised by both the reviewers and they have added a data availability statement as recommended. The manuscript meets formatting specifications

7. PLOS authors have the option to publish the peer review history of their article (what does this mean?). If published, this will include your full peer review and any attached files.

Reviewer #3: **Yes: **Oscar Mbembela

Reviewer #4: **Yes: **Fredirick Mashili

---

## [Editor Report · Acceptance letter]

13 Feb 2024

PONE-D-23-02201R2 

PLOS ONE

Dear Dr. Nasreddine, 

I'm pleased to inform you that your manuscript has been deemed suitable for publication in PLOS ONE. Congratulations! Your manuscript is now being handed over to our production team.

Kind regards, 

on behalf of

Dr Fredirick Lazaro mashili 

Academic Editor

PLOS ONE